Corrected: Author correction; Author correction

# Efficient electron transfer across hydrogen bond interfaces by proton-coupled and -uncoupled pathways

Tao Cheng[1], Dong Xue Shen[1], Miao Meng[1], Suman Mallick[1], Lijiu Cao[1], Nathan J. Patmore [2], Hong Li Zhang[1], Shan Feng Zou[1], Huo Wen Chen[1], Yi Qin[1], Yi Yang Wu[1] & Chun Y. Liu [1]

Thermal electron transfer through hydrogen bonds remains largely unexplored. Here we report the study of electron transfer through amide-amide hydrogen bonded interfaces in mixed-valence complexes with covalently bonded $Mo_2$ units as the electron donor and acceptor. The rate constants for electron transfer through the dual hydrogen bonds across a distance of 12.5 Å are on the order of $\sim 10^{10}$ s$^{-1}$, as determined by optical analysis based on Marcus–Hush theory and simulation of $\nu$(NH) vibrational band broadening, with the electron transfer efficiencies comparable to that of $\pi$ conjugated bridges. This work demonstrates that electron transfer across a hydrogen bond may proceed via the known proton-coupled pathway, as well as an overlooked proton-uncoupled pathway that does not involve proton transfer. A mechanistic switch between the two pathways can be achieved by manipulation of the strengths of electronic coupling and hydrogen bonding. The knowledge of the non-proton coupled pathway has shed light on charge and energy transport in biological systems.

[1] Department of Chemistry, Jinan University, 601 Huang-Pu Avenue West, 510632 Guangzhou, China. [2] Department of Chemical Sciences, University of Huddersfield, Queensgate, Huddersfield HD1 3DH, UK. Correspondence and requests for materials should be addressed to C.Y.L. (email: tcyliu@jnu.edu.cn)

Understanding electronic coupling (EC) and electron transfer (ET) across hydrogen bonds (HBs) is of fundamental importance in elucidating important biochemical processes in a diverse number of biological systems, including enzymes, proteins, and DNA[1]. Unlike covalent bonds, hydrogen-bonding interactions (X–H⋯Y) are predominantly electrostatic in nature, comprising a single-electron owned H atom in the middle of three linearly connected atoms, which makes an HB relatively weak, flexible, and dynamic. The process of transferring electrons across such an intermolecular interface has long inspired explorations of theoretical and experimental chemists[2,3]. In this context, the two major questions are (i) how efficiently are H bonds able to transport electrons and (ii) whether the ET process occurs with the help of the proton. The first quantitative evaluation of through-HB ET by comparison with covalent σ and π bonds employed an elegantly designed D–HB–A molecule with zinc(II) and iron(III) porphyrins as the electron donor (D) and acceptor (A), respectively. Charge transfer across this acid–acid HB interface in the photoexcited states was monitored by ultrafast transient spectroscopy[4]. The study led to a remarkable conclusion that the extent of EC through HBs is greater than via C–C σ bonds, but lower than across a π-conjugated bridge. However, significant discrepancies have arisen from subsequent experimental[5] and theoretical[6] studies, and thus the efficiency of EC mediation by an HB interface, relative to covalent bonds, remains controversial.

Donor–acceptor ET crossing an HB interface is often accompanied by a proton transfer reaction associated with two HB states, X–H⋯Y (initial) and X⋯H–Y (final); the overall ET reaction is referred to as proton-coupled electron transfer (PCET). Therefore, X–H bond breakage and Y–H bond formation are the prerequisites for a PCET reaction to be considered. Within the classical ET theoretical framework, contributions from Cukier and Nocera[7] and Hammes–Schiffer and Stuchebrukhov[8] have led to the development of PCET theories by reformulating the Fermi's golden rule with a modified Frank–Condon factor. To account for the large displacement of the proton, the proton motions are further separated from other nuclear vibrations in the spirit of the Born-Oppenheimer approach[9]; by doing so, the EC element for the unperturbed diabatic states is redefined[7,8]. Nonadiabatic PCET theories treat ET and PT as separate particle tunneling events, which may occur sequentially or in concert. Unfortunately, employing the golden rule formalism to derive ET rate constants necessitates the knowledge of some physical parameters that are not always experimentally available, which prevents validation of the theories and their broad application. In a D–HB–A system, vibronic coupling plays an important role in determining the ET kinetics due to the dynamic nature of the bridge. To address this issue, one needs to scale the HB strength, and thus, topologically well-defined experimental models are in demand. Furthermore, up to now, experimental studies of PCET kinetics has been focused on photoinduced ET[7], which feature nonadiabaticity and high exothermicity. However, a number of biochemical systems undergo PCET without light and under less exothermic conditions; typical examples include light-independent reactions in the Calvin cycle and cellular respiration catalyzed by cytochrome c oxidase[1,10].

Mixed-valence (MV) D–B–A molecules, which have identical D and A sites differing only in formal oxidation states, are valuable experimental models for study of electron self-exchange reaction with Marcus–Hush theory[11,12], which has been successfully used to evaluate intramolecular ET through covalently bonded bridges[13,14]. Recent efforts have produced various examples of hydrogen bonded MV complexes, in which self-complementary HB interactions are used to bridge the electron donor and acceptor[15–17], with the aim to evaluate the D–A EC optically upon analysis of the intervalence charge transfer (IVCT) absorbance. However, this optical feature has only been observed in few examples of hydrogen bonded MV compounds, and lack of a test-bed series of compounds has hindered the kinetic study of thermal ET in electron self-exchange reactions with zero driving force ($-\Delta G^{\circ} = 0$)[5,16,17].

Herein we report the study of thermally induced ET across an HB bridge in symmetrical MV D–HB–A systems ($-\Delta G^{\circ} = 0$) with a quadruply bonded $Mo_2$ unit as the donor, and a $Mo_2$ unit with a bond order of 3.5 as the acceptor. The quadruply bonded $Mo_2$ unit has an electronic configuration of $\sigma^2\pi^4\delta^2$[18], and in a $Mo_2$ D–B–A system, the δ orbital in the $Mo_2$ unit is discriminated from the σ and π orbitals by symmetry and energy. Therefore, only the δ electrons in the HOMO of the $Mo_2$ core are involved in ET from the donor to acceptor, which makes the system unique and most desirable as an experimental model for study of EC and ET[14,19]. By analysis of IVCT absorptions for the $Mo_2$ D–HB–A systems reported here, we were able to determine the EC matrix elements ($H_{ab}$) from the Mulliken–Hush expression[20] and study the ET kinetics by employing Marcus–Hush theory[11,12,20], which provides results in excellent agreement with the data obtained from analysis of ν(NH) vibrational band broadening. Comparisons of ET kinetics with the data obtained from analogous $Mo_2$ dimers having π (phenylene) and σ (cyclohexylene) bond bridges that span similar D–A distances show that the HB interface conducts EC and ET as well as the π-conjugated bridges. This study has validated the PCET theory for electron self-exchange reactions, and demonstrated a usually overlooked, proton uncoupled electron transfer (PUET) pathway of thermal electron transfer in HB systems.

## Results

**Characterization of the HB bridged $Mo_2$ dimers.** The three categories of $Mo_2$ complexes studied are of a general formula [$Mo_2$]–bridge–[$Mo_2$], in which [$Mo_2$] = [$Mo_2(DArF)_3(O_2C)$] (DArF = $N,N'$- diarylformamidinate) and the bridges are an amide–amide HB (**1**), phenylene (**2**), or cyclohexylene (**3**), as shown in Fig. 1a. To fine-tune the EC of the system[21], each series consists of four members which are differentiated by the *para* substituents X (X = $N(CH_3)_2$ (**a**), $CH(CH_3)_2$ (**b**), $OCH_3$ (**c**), or $CH_3$ (**d**)) on the DArF ligands. The HB bridged $Mo_2$ dimers (**1a–d**) are self-assembled in less polar solvents from two $Mo_2$ paddlewheel monomers, [$Mo_2(DArF)_3(O_2CCONH_2)$], in which the fourth equatorial coordination position of the $Mo_2$ center is occupied by an acetamide ligand.

The four HB-assembled $Mo_2$ dimers (**1a–d**) have been structurally characterized by X-ray diffraction. The crystal structures are presented in Fig. 1b and the selected bond parameters are listed in Table 1. The structures of **1a–d** show that the $Mo_2$ monomers are dimerized in solid state through amide–amide dual HBs, giving these dimers the same topological geometry as that of the covalent π (**2a–d**) and σ bond (**3a–d**) bridged analogs (Fig. 1a). The Mo–Mo bond distances are about 2.1 Å, which is comparable to quadruple bonds with similar supporting ligands[21], and the $Mo_2$⋯$Mo_2$ separations of ∼12.5 Å are slightly longer than those for series **2** and **3** (∼11.25 Å)[21]. The HB bridge is built by a pair of intermolecular N–H⋯O hydrogen bonds with N⋯O distances of 2.865 Å (**1b**) < 2.878 Å (**1d**) < 2.885 Å (**1a**) < 2.976 Å (**1c**) and ∠N–H⋯O angles of 172.46° (**1a**) > 169.96° (**1d**) > 154.42° (**1b**) > 151.35° (**1c**). A $sp^2$ N atom and N–H bond distance of 0.86 Å

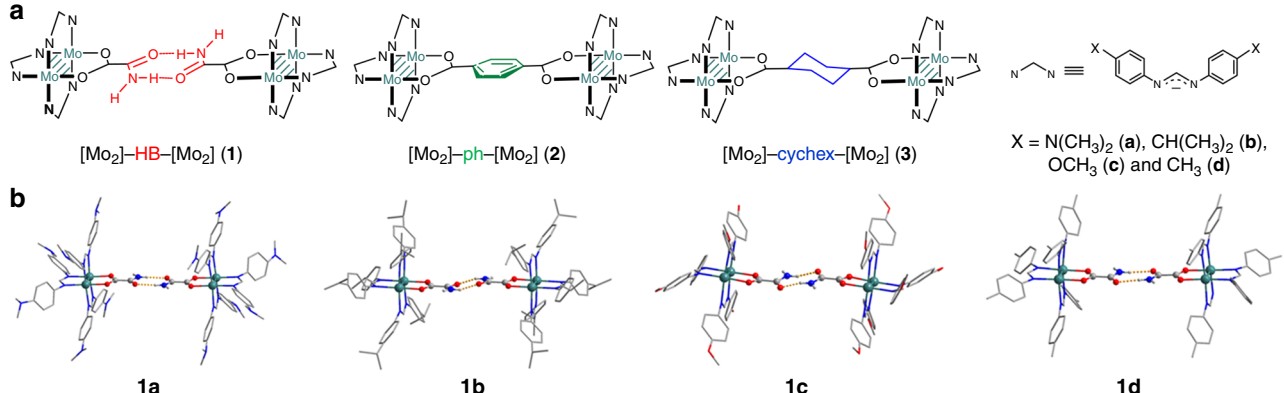

**Fig. 1 a** Molecular skeletons for [Mo$_2$]–bridge–[Mo$_2$] complexes under investigation. **b** X-ray crystal structures for the Mo$_2$ dimers bridged by amide–amide dual hydrogen bonds (H atoms are omitted for clarity). Each series (**1–3**) consists of four Mo$_2$ dimers that differ in the substituents (X, **a-d**) on the formamidinate auxiliary ligands. By varying the electron-donating ability of X, the donor–acceptor electronic coupling in the mixed-valence complexes is tuned (see text)

**Table 1 Selected distances (Å) and angles (°) for 1a–d**

|  | 1a·7CH$_2$Cl$_2$ | 1b·7CH$_2$Cl$_2$ | 1c | 1d·CH$_2$Cl$_2$ |
|---|---|---|---|---|
| Mo(1)–Mo(2) | 2.0921(4) | 2.095(1) | 2.0909(3) | 2.0919(4) |
| Mo(1)–O(7) | 2.156(3) | 2.162(9) | 2.136(2) | 2.144(2) |
| Mo(2)–O(8) | 2.143(3) | 2.120(9) | 2.162(2) | 2.147(2) |
| C(1)–C(2) | 1.522(6) | 1.54(2) | 1.534(5) | 1.514(5) |
| Mo$_2$···Mo$_2$ | 12.588 | 12.455 | 12.498 | 12.571 |
| C(2)–C(2a) | 4.064 | 3.983 | 4.094 | 4.062 |
| N–$H$···O | 2.885 | 2.865 | 2.976 | 2.878 |
| H···O | 2.031 | 2.066 | 2.174 | 2.025 |
| ∠N–H···O | 172.46 | 154.42 | 151.35 | 169.96 |

are assumed for the amide group in all these compounds, resulting in O···H distances of 2.025 Å (**1d**) < 2.031 Å (**1a**) < 2.066 Å (**1b**) < 2.174 Å (**1c**). Therefore, the structural parameters show that the HBs in **1a** and **1d** are appreciably stronger than those in **1b** and **1c**. It appears that **1a** has strong HBs, which may be due to the strong electron donating of the N(CH$_3$)$_2$ groups, but there is no strict correlation between HB strength and the electronic property of the X substituents. The HBs in these adducts are generally weak, in comparison with distances of <2.65 Å expected for strong N–H···O bonds[22]. However, the dual HBs in the amide–amide linkage should strengthen the bonding between the two Mo$_2$ units. The $^1$H NMR spectra of compounds **1a–d** in CDCl$_3$ exhibit two widely separated resonances for the amide protons (N–$H$) at ~5.6 ppm for the free proton and at ~7.2 ppm for the bonded proton, a small downfield chemical shift indicating relatively weak HBs[22]. In contrast, in deuterated dimethylsulfoxide (DMSO-$d_6$), the two amide protons exhibit similar chemical shifts (~8 ppm), corresponding to the monomeric precursor; therefore, the dimeric structures of **1a–d** in solid state and in less polar solvents such as dichloromethane (DCM) are clearly established.

Compounds **1b–d** show two closely separated redox couples in their cyclic voltammograms (CVs) in DCM (Supplementary Figure 2). The chemical potentials ($E_{1/2}$) fall in the range of −0.3 to 0.1 V (vs. Fc$^{+/0}$) and the potential separations ($\Delta E_{1/2}$) are estimated to be ~100 mV from Richard–Taube methods (Supplementary Figure 31)[23], as listed in Table 2. In DMF, these complexes show only one redox couple, as expected, for

the corresponding monomers (Supplementary Figure 3)[15]. For **1a**, **2a**, and **3a**, the N(CH$_3$)$_2$ groups on the DArF ancillary ligands are redox active ($E_{1/2}$ = 0.1–0.5 V vs. Fc$^{+/0}$), which makes the redox waves of the Mo$_2$ centers weak and irreversible. For the other HB adducts (**1b**, **1c**, and **1d**), the redox processes are attributed to the two one-electron oxidations occurring on each of the Mo$_2$ centers, and the $E_{1/2}$ and $\Delta E_{1/2}$ values are comparable with the phenylene bridged dimers. It is interesting to note that the $\Delta E_{1/2}$ values for **1b–d** are larger in most cases than those for the corresponding phenylene bridged analogs (**2b–d**) (Table 2 and Supplementary Figure 32). This is especially the case for **1c**, which is notable given that the HB bridge separates the two Mo$_2$ units even further than the phenylene bridge in **2c** (Supplementary Figure 4). The electrochemical behaviors demonstrate that these Mo$_2$ complexes exist as HB dimers in both neutral and the oxidized states in DCM solution. This is contrary to the Ru$_3$O D–HB–A system in which the HB bridged dimer is formed upon reduction of one of the metal cluster to yield the mixed-valent species[24]. The cyclohexylene bridged complexes **3a–d** present the smallest $\Delta E_{1/2}$ values (~70 mV) (Supplementary Figures 5 and 33), showing very weak coupling due to the electrostatic effect. The increased magnitude of $\Delta E_{1/2}$ for **1a–d** account for the large resonant effect on the donor–acceptor coupling (vide infra) and probably the dynamic properties of the HB adducts as well.

All the HB adducts in DCM exhibit two isolated IR bands at ~3517 and at ~3398 cm$^{-1}$, labeled as $v$(NH)$_A$ and $v$(NH)$_B$ in Fig. 2a–d, respectively. Similar $v$(NH) bands are observed at 3529 and 3319 cm$^{-1}$ for the 2-aminopyridine dimer[25,26], and at 3364 and 3166 cm$^{-1}$ for the benzamide dimers[27]. Evidently, $v$(NH)$_A$ and $v$(NH)$_B$ result from stretching vibrations of the free and bonded N–H bonds in the amide–amide dual HB linkage, respectively. The appearance of only two $v$(NH) bands in the spectrum signals the presence of an inversion center on the HB motif. In DMF, two $v$(NH) bands with similar stretching frequencies, ~3545 and ~3485 cm$^{-1}$, are observed, indicating the presence of Mo$_2$ monomers (Supplementary Figure 1)[25]. It is important to note that **1a** and **1d** exhibit an extra band at 3613 cm$^{-1}$(Fig. 2a, d) that is absent for **1b** and **1c**. This high frequency band is assigned to the O–H stretching mode and is a direct evidence for tautomerism in these two systems[28]. The appearance of $v$(OH) band in the spectra of **1a** and **1d** is ascribable to the stronger HBs that lower the PT energy barrier[22].

**Table 2 Summary of electrochemical data, IVCT band parameters, electronic coupling constants, and ET kinetics for the mixed-valence complexes (1a-d)$^+$ and (2a-d)$^+$**

| System | X | $\Delta E_{1/2}$ (mV) | $E_{IT}$ ($\lambda$) (cm$^{-1}$) | $\varepsilon_{IT}$ (M$^{-1}$ cm$^{-1}$) | $\Delta\nu_{1/2}$ (exp) (cm$^{-1}$) | $H_{ab}$ (cm$^{-1}$) | $\Delta G^\star$ (cm$^{-1}$) | $k_{adia}$ (s$^{-1}$) | $k_{nonadia}$ (s$^{-1}$) | $k_{obs}$ (s$^{-1}$) |
|---|---|---|---|---|---|---|---|---|---|---|
| 1a | N(CH$_3$)$_2$ | – | 4000 | 1769 | 2835 | 410 | 632 | $2.4 \times 10^{11}$ | $8.0 \times 10^{10}$ a | $6.0 \times 10^{10}$ |
| 1b | CH(CH$_3$)$_2$ | 102 | 4253 | 564 | 1873 | 194 | 878 | $7.2 \times 10^{10}$ | $7.9 \times 10^{10}$ | $2.0 \times 10^{10}$ |
| 1c | OCH$_3$ | 138 | 4607 | 385 | 1982 | 172 | 986 | $4.3 \times 10^{10}$ | $3.9 \times 10^{10}$ | $3.0 \times 10^{10}$ |
| 1d | CH$_3$ | 110 | 4064 | 231 | 2132 | 130 | 890 | $6.8 \times 10^{10}$ | $4.5 \times 10^{10}$ | $5.0 \times 10^{10}$ |
| 2a | N(CH$_3$)$_2$ | – | – | – | – | – | – | – | – | – |
| 2b | CH(CH$_3$)$_2$ | 95 | 4506 | 869 | 4362 | 460 | 713 | $1.6 \times 10^{11}$ | $3.2 \times 10^{11}$ | – |
| 2c | OCH$_3$ | 91 | 4633 | 755 | 4456 | 440 | 760 | $1.3 \times 10^{11}$ | $2.5 \times 10^{11}$ | – |
| 2d | CH$_3$ | 112 | 4871 | 629 | 4557 | 416 | 837 | $8.8 \times 10^{10}$ | $1.6 \times 10^{11}$ | – |

For **3a–d**, electrochemical data are shown in Supplementary Figure 33 and spectroscopic data are shown in Supplementary Figure 42
a For **1a$^+$**, $k_{nonadia} = k_{PCET}$, determined by $(k_{obs})^{-1} = (k_{ET})^{-1} + (k_{PCET})^{-1}$ where $k_{ET} = k_{adia}$

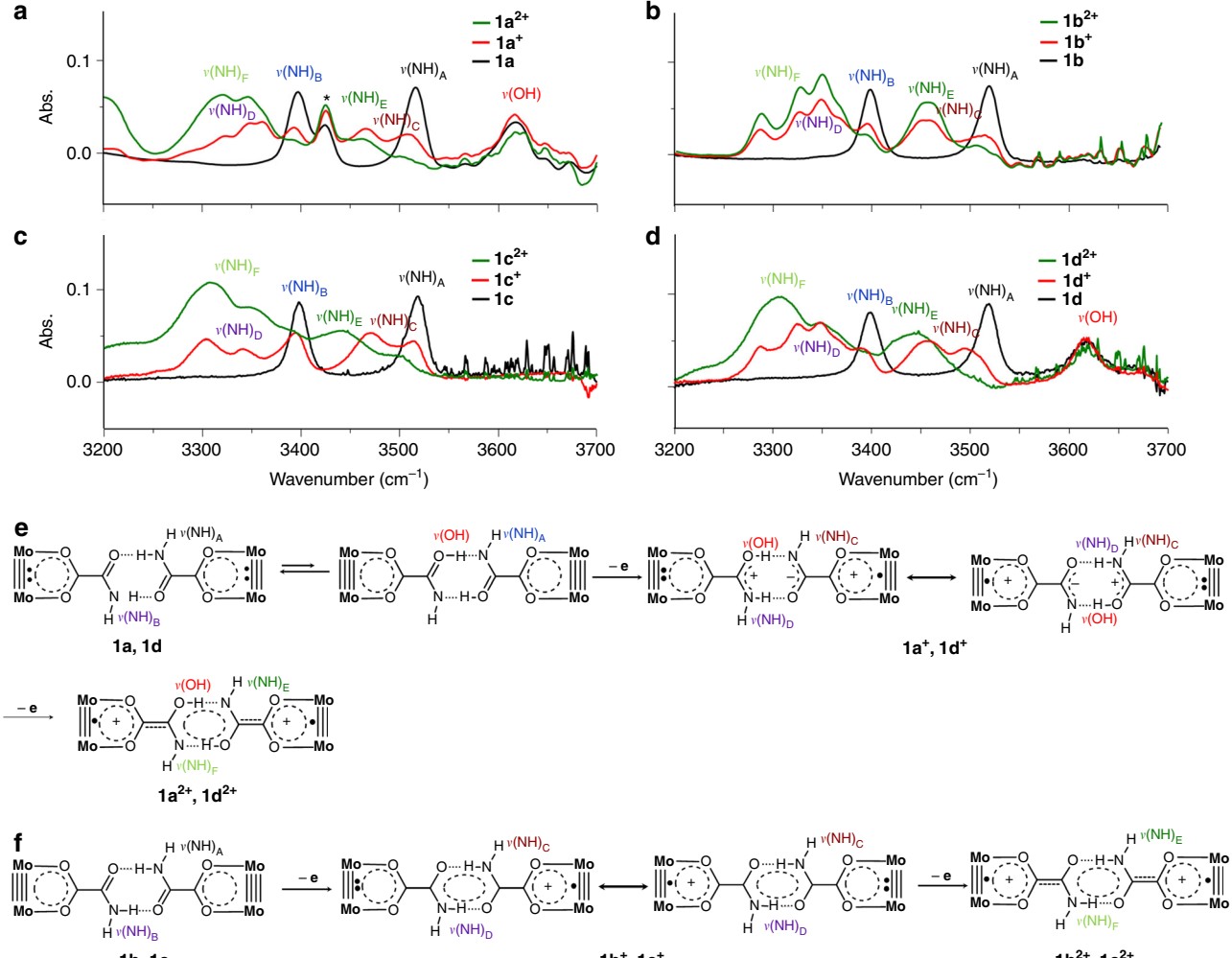

**Fig. 2** Infrared spectra and assignments of N–H vibrations for **1**, **1$^+$**, and **1$^{2+}$**. **a** X = N(CH$_3$)$_2$. **b** X = CH(CH$_3$)$_2$. **c** X = OCH$_3$. **d** X = CH$_3$. **e** Assignments of the IR N–H and O–H stretching bands for **1a** and **1d** in different oxidation states (0, +1 and +2). **f** Assignments of the IR N–H stretching bands for **1b** and **1c** in different oxidation states (0, +1 and +2). Note that a $\nu$(OH) band at 3613 cm$^{-1}$ is observed for systems **1a** and **1d**, but not for **1b** and **1c**. For the system with X = N(CH$_3$)$_2$, the 3425 cm$^{-1}$ band (marked with an asterisk), which is not affected by the ET and PT processes, is a frequency double band for a mid-IR band (1713 cm$^{-1}$)

Contrarily, the non-linear structure (Fig. 1b, c) of weaker HBs in **1b** and **1c** inhibits the proton transfer.

The cationic mixed-valence complexes of the three series (**1$^+$**, **2$^+$**, and **3$^+$**) were prepared by one-electron oxidation of the

corresponding neutral precursors using a stoichiometric amount of the oxidizing reagent ferrocenium hexafluorophosphate. The electron paramagnetic resonant (EPR) and UV–Vis–NIR spectra are recorded in situ. In the EPR spectra, all the MV complexes

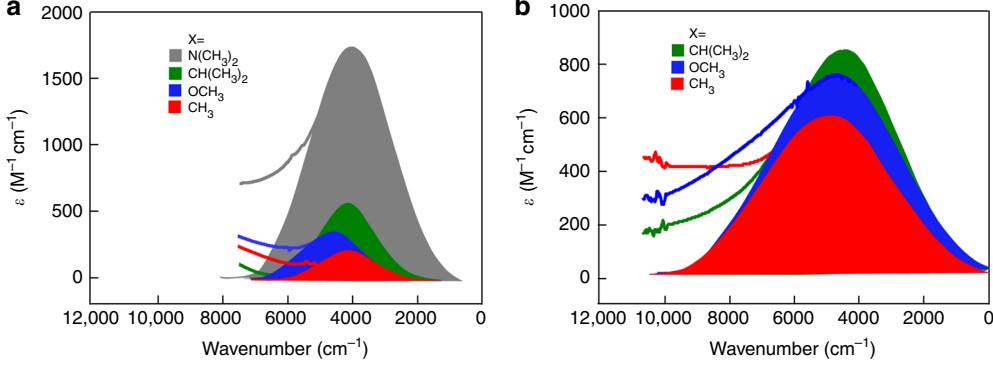

**Fig. 3** Near- to mid-infrared spectra of the mixed-valence complexes showing the vibronic intervalence charge transfer absorption bands. **a** For the hydrogen bond bridged MV complexes (**1a–d**)$^+$ and **b** for the phenylene bridged MV complexes (**2b–d**)$^+$. For complexes (**1a–d**)$^+$ and (**2b–d**)$^+$ a symmetrical IVCT absorption band is observed. The same color code is used to show the full Gaussian-shaped IVCT band profiles for differerent MV systems with the same X substituent

exhibit a symmetric peak at $g = \sim1.94$ (Supplementary Figure 6), indicating that for all Mo$_2$ the dimers, the unpaired electron is mainly localized in the δ orbital[29,30]. The striking optical feature observed in the spectra for the MV complexes **1**$^+$ (Supplementary Figures 34–37) and **2**$^+$ (Supplementary Figures 39–41), except for **2a**$^+$, is a broad, symmetrical absorption band in the near-IR region, which is attributed to IVCT between the two Mo$_2$ centers, as shown in Fig. 3. For both of the systems, the IVCT band parameters, including transition energy ($E_{IT}$), band intensity ($\varepsilon_{IT}$), and half bandwidth ($\Delta v_{1/2}$), are comparable with those found for other Mo$_2$ dimers with various π bridges differing in length[14,31,32], symmetry[19], conjugation[33], and conformation[34]. As expected, the absorption spectra of the corresponding monomers, [Mo$_2$(DArF)$_3$(O$_2$CCONH$_2$)]$^{0/+}$, obtained in DMF do not exhibit the characteristic IVCT transition, as shown in Supplementary Figure 38. Therefore, the presence of a well-defined IVCT band for each of the HB adducts confirms their dimeric structure and the mixed-valency of the singly oxidized complexes. By contrast, no IVCT absorbance was observed in the spectra of complexes **3a**$^+$-**d**$^+$ (Supplementary Figure 42), which indicates very weak EC between the two bridged Mo$_2$ centers in this series, consistent with the electrochemical results.

Notably, for **1**$^+$, the intervalence transition energies ($E_{IT}$) in the range of 4000–4500 cm$^{-1}$ are considerably lower than those for **2**$^+$ ($E_{IT}$ = 4500–4850 cm$^{-1}$), and the bandwidths ($\Delta v_{1/2}$) are only about a half of the $\Delta v_{1/2}$ values for **2**$^+$ (Fig. 3, Supplementary Figure 8 and Table 2). These results are consistent with the large $\Delta E_{1/2}$ values observed for **1a–d**, as a low energy, narrow IVCT band indicates strong EC. As expected, the strong electron donating of the substituents in **1a**$^+$ and **2b**$^+$ gives rise to the lowest transition energy $E_{IT}$ in the series (Table 2). However, for **1**$^+$, the IVCT absorption intensities ($\varepsilon_{IT}$, M$^{-1}$ cm$^{-1}$) are generally lower than those for **2**$^+$, except for **1a**$^+$ that exhibits an exceptionally intense IVCT band (Fig. 3 and Table 2). Unlike other members of the **2**$^+$ series, **2a**$^+$ does not exhibit an IVCT band. This different optical behavior can be rationalized by the redox active N(CH$_3$)$_2$ groups, which act as electron donors and become the redox partner of the cationic Mo$_2$ unit. Indeed, for **2a**$^+$, an extra charge transfer absorption band is observed at $\lambda = 1250$ nm, which is tentatively assigned to charge transfer from the dimethylamine N atom to the Mo$_2$ center.

**Determination of $H_{ab}$ and $k_{ET}$ data for the Mo$_2$ dimers.** The well-defined IVCT bands for **1**$^+$ and **2**$^+$ allow determination of the electronic coupling matrix elements $H_{ab}$ from the

Mulliken–Hush expression (Eq. (1)) [12,20].

$$H_{ab} = 2.06 \times 10^{-2} \frac{\sqrt{E_{IT} \varepsilon_{IT} \Delta v_{1/2}}}{r_{ab}}. \tag{1}$$

Considering that the δ electrons are delocalized over the coordination shell of the Mo$_2$ center[14,19], the effective ET distances ($r_{ab}$) were estimated from the C to C distance between the two linked carboxylic groups, and found to be $\sim7.11$ Å for **1**$^+$ and $\sim5.85$ Å for **2**$^+$. These effective ET distances are around half of the Mo$_2$···Mo$_2$ separations, close to 40% of the geometrical distance between donor and acceptor, in accordance with previous literature reports[35]. For the **1**$^+$ series, the largest coupling parameter, $H_{ab} = 410$ cm$^{-1}$, is found for **1a**$^+$; while for the other three adducts, the $H_{ab}$ values decrease from 194 to 130 cm$^{-1}$ as the substituents become less electron donating (Table 2). For all the complexes in series **2**$^+$, except for **2a**$^+$, similar coupling constants, $H_{ab} \approx 450$ cm$^{-1}$, are found. Therefore, the MV compounds in series **1**$^+$ and **2**$^+$ belong to Class II in Robin–Day's scheme[36]. For both series, the magnitudes of $H_{ab}$ vary following the same trend, showing remarkable correlation to the remote substituents as observed in previous work[21]. Interestingly, it appears that EC through an HB interface is even more sensitive to the electronic property of the substituents than through a π bridge (Table 2).

The activation energy $\Delta G^\star$ for system crossing the transition state of ET reaction is calculated according to Marcus theory (Eq. (2)) [11,37],

$$\Delta G^* = \frac{(\lambda - 2H_{ab})^2}{4\lambda}, \tag{2}$$

where reorganization energy $\lambda = E_{IT} + \Delta G^\circ$ and thus, for the current ET systems with a free energy change of zero ($\Delta G^\circ = 0$), $\lambda = E_{IT}$[12,20,37]. For **1a**$^+$ and **2**$^+$, the calculated $\Delta G^\star$ (Table 2) are substantially smaller than $\lambda/4$, the $\Delta G^\star$ introduced by Marcus[38] for the nonadiabatic limit of ET reactions, but close to $\lambda/4$ for **1b**$^+$, **1c**$^+$, and **1d**$^+$. This means that **1a**$^+$ and the three phenylene bridged analogs (**2**$^+$) should be treated adiabatically, whereas the three weakly coupled D–HB–A systems are essentially on the border of the adiabatic and nonadiabatic regimes. This is further justified by the magnitude of $2H_{ab}$, the separation between the adiabatic potential energy surfaces[12]. For **1a**$^+$ and **2**$^+$, $2H_{ab}$ are 4–5 times larger than the thermal energy barrier, $k_BT$ (207 cm$^{-1}$),[9] while for **1b**$^+$, **1c**$^+$ and **1d**$^+$, $2H_{ab} \sim k_BT$. The electron hopping frequencies ($v_{el}$) are calculated in the range $10^{13}$–$10^{14}$ s$^{-1}$, close to nuclear vibrational frequency ($v_n$) of $10^{12}$–$10^{13}$ s$^{-1}$ [37]. For the HB systems, $v_n$ is reduced by the

nonadiabaticity of PT. Therefore, for $\mathbf{1b^+}$–$\mathbf{d^+}$, the ET kinetics can be described either adiabatically using the Arrhenius equation (Eq. (3)) with an pre-exponential factor ($A$) of $5 \times 10^{12}\,\text{s}^{-1}$ [37], or non-adiabatically, using the rate expression (Eq. (4)) developed by Levich[39], while for $\mathbf{1a^+}$ and $\mathbf{2^+}$, adiabatic treatment is more appropriate.

$$k_{\text{ET}} = A \exp\left(-\frac{\Delta G^*}{kT}\right), \qquad (3)$$

$$k_{\text{ET}} = \frac{2H_{ab}^2}{h}\sqrt{\frac{\pi^3}{\lambda kT}}\exp\left(-\frac{\lambda}{4kT}\right). \qquad (4)$$

For the three weakly coupled HB bridged dimers ($\mathbf{1b^+}$–$\mathbf{d^+}$), almost identical $k_{\text{adia}}$ and $k_{\text{nonadia}}$ are obtained (Table 2), as expected, and the calculated rate constants in the range of $4$–$8 \times 10^{10}\,\text{s}^{-1}$ show a general substituent-dependence of $k_{\text{ET}}$, that is, that electron-donating of the substituents (X) accelerates ET. For $\mathbf{1a^+}$ and $\mathbf{2^+}$, the two approaches give the rate constants of $\sim 10^{11}\,\text{s}^{-1}$, with $k_{\text{adia}} < k_{\text{nonadia}}$ by a factor of two (Table 2) and the rate constants are larger than those for $\mathbf{1b^+}$–$\mathbf{d^+}$ by one order of magnitude or less.

For the D–HB–A systems, we are able to measure the ET rate constants by analysis of $\nu$(NH) band broadening in the IR spectra, a method used for determination of ET rate constants in $\text{Ru}_3\text{O}$–$\text{Ru}_3\text{O}$ MV systems[24,40]. This methodology has also been employed to evaluate thermally induced ET reactions in MV D–B–A systems with covalent bridges using NMR[41] and EPR[13,35,42] spectroscopy techniques. In the IR spectra of $\mathbf{1^+}$, substantial changes of $\nu$(NH) frequency and band shape are observed, in comparison with the spectra for $\mathbf{1}$. As shown in Fig. 2a–d, the MV species ($\mathbf{1^+}$) show partially coalesced bands, $\nu$(NH)$_\text{C}$, for the free N-H bonds, while the bonded N-H bonds exhibit multiple absorptions shifted towards lower energy, denoted as $\nu$(NH)$_\text{D}$. The manifolds of the N–H vibrations show directly the multiple nuclear degrees of freedom caused by the ET or PCET. For the dication complexes ($\mathbf{1a^{2+}}$–$\mathbf{d^{2+}}$) obtained by two-electron oxidation, the free N–H bonds present a broad and

intense band at $\sim 3460\,\text{cm}^{-1}$ ($\nu$(NH)$_\text{E}$) vs. $3517\,\text{cm}^{-1}$ ($\nu$(NH)$_\text{A}$) for the neutral complexes $\mathbf{1a}$–$\mathbf{d}$. It is worthwhile to note that the IR spectra of $\mathbf{1b^{2+}}$ and $\mathbf{1c^{2+}}$ exhibit a small shoulder at higher energy ($\sim 3460\,\text{cm}^{-1}$). This feature arises from disproportionation equilibrium of the complexes, indicating poor stabilization of the MV state in $\mathbf{1b^+}$ and $\mathbf{1c^+}$ due to weak EC[43]. Therefore, for the MV species ($\mathbf{1^+}$), the coalesced $\nu$(NH)$_\text{C}$ band profile is reflective of the dynamics of the free N–H bonds caused by ET across the HB interface (Fig. 2a–d), and thus can be exploited to simulate the ET kinetics[24,40]. For each of the four adducts band simulations were performed using the published software Zoerbex[44], as shown in Fig. 4, giving the observed rate constants ($k_{\text{obs}}$) listed in Table 2. Remarkably, the $k_{\text{obs}}$ values are in excellent agreement with the calculated rate constants, except for $\mathbf{1a^+}$. In the series, the largest rate constant, $k_{\text{obs}} = 6.0 \times 10^{10}\,\text{s}^{-1}$, is obtained for $\mathbf{1a^+}$ and the smallest rate, $k_{\text{obs}} = 2.0 \times 10^{10}\,\text{s}^{-1}$ for $\mathbf{1b^+}$, essentially following the same trend in the calculated rate constants. Recently, Kubiak's group reported a rate constant of $3.8 \times 10^{11}\,\text{s}^{-1}$ derived by broadening analysis of the $\nu$(CO) band for an HB bridged $\text{Ru}_3\text{O}$–$\text{Ru}_3\text{O}$ MV complex, which has shorter ET distance[24]. Comparison of ET rates for these two distinct systems verifies the magnitudes of $k_{\text{ET}}$ in the $\text{Mo}_2$–$\text{Mo}_2$ system. It is noted that for $\mathbf{1a^+}$ the deviation of $k_{\text{obs}}$ ($6.0 \times 10^{10}\,\text{s}^{-1}$) from $k_{\text{adia}}$ ($2.4 \times 10^{11}\,\text{s}^{-1}$) is relatively large. This disagreement can be rationalized by the involvement of two ET reaction channels, i.e. ET and PCET[2]. While thermal ET is controlled by the transition state through EC ($H_{ab}$), nonadiabatic PCET is governed by vibronic coupling ($S$)[42,45]. For $\mathbf{1a^+}$ with $2H_{ab} \gg k_{\text{B}}T$, it is inappropriate to calculate the rate constant using $H_{ab}$ for the PCET reaction in the nonadiabatic regime. From this point of view, the PCET rate constant, namely, $k_{\text{PCET}}$, can be derived from Eq. (5) [2].

$$\frac{1}{k_{\text{obs}}} = \frac{1}{k_{\text{ET}}} + \frac{1}{k_{\text{PCET}}}, \qquad (5)$$

where $k_{\text{ET}} = k_{\text{adia}}$ and $k_{\text{PCET}} = k_{\text{nonadia}}$, viewing the concerted PCET process as a single tunneling event that crosses a longer

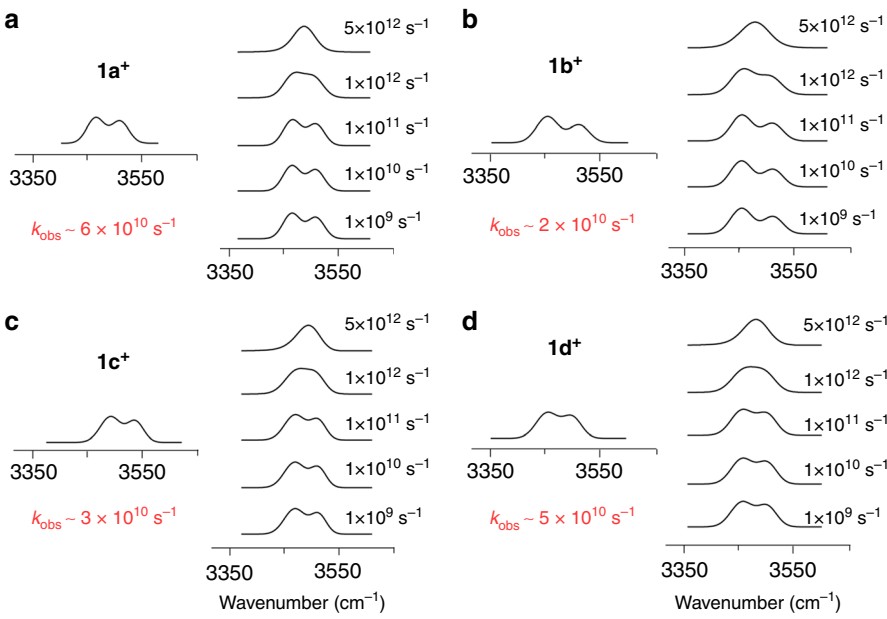

**Fig. 4** Comparison of observed (left) and simulated (right) stretching frequencies for the free N–H bonds in the MV complexes $\mathbf{1a^+}$ (**a**, X = N(CH$_3$)$_2$), $\mathbf{1b^+}$ (**b**, X = CH(CH$_3$)$_2$), $\mathbf{1c^+}$ (**c**, X = OCH$_3$), and $\mathbf{1d^+}$ (**d**, X = CH$_3$). Note that in the simulated spectra (right), the degree of band coalesce varies as a function of the intermolecular electron transfer rate. For each of the complex systems, the rate constant ($k_{\text{obs}}$) is determined by comparison of the spectrum with the simulated spectra (Supplementary Figures 43−46 and Supplementary Table 2)

tunneling path over a higher energy barrier. Indeed, Eq. (5) gives $k_{PCET} = 8.0 \times 10^{10}$ s$^{-1}$ for $\mathbf{1a}^+$, in good agreement with $k_{obs}$ of $6.0 \times 10^{10}$ s$^{-1}$ (Table 2). In other words, the nonadiabaticity caused by PT lowers the rate constant by less than one order of magnitude, which is in agreement with the consideration of nonadiabaticity in PCET theories[7,8], The ET rate constants in these Mo$_2$ MV systems are comparable with the data obtained for photoinduced ET across the acid–acid HB interface in metal porphyrin-based donor–acceptor systems, and thus verify the order of capability of the bridges in modulating EC and ET, that is, C–C π bonds > HBs > C–C σ bonds[4]. Moreover, this MV series provides an unprecedented example showing the transition of ET systems from the vibronically adiabatic regime ($\mathbf{1a}^+$) to the nonadiabatic regime ($\mathbf{1b}^+$, $\mathbf{1c}^+$, and $\mathbf{1d}^+$)[45], modulated by EC. This study substantiates the semiclassical theory with excellent consistency between the calculated and experimental data, which is a long-term pursuit in the field of ET[5]. To our knowledge, this is the first example that compares optical and thermal ET pathways for systems on the adiabatic/nonadiabatic borderline[45] and on the vibrational time scale ($\sim 10^{-12}$ s), whereas earlier works have verified the related theories in the adiabatic[13,42] and nonadiabatic[41] limits on the time scales of NMR ($\sim 10^{-6}$ s) and EPR ($\sim 10^{-8}$ s).

## Discussion

The agreement between the measured $k_{obs}$ and the calculated $k_{ET}$ for these D–HB–A systems proves the accuracy of the optically determined rate constants for electron self-exchange in the phenylene bridged MV Mo$_2$ dimers. It is found that the rate constants ($k_{adia}$) and ET distances ($R_{ab}$) of $\mathbf{1c}^+$ and $\mathbf{2c}^+$ fit well the linear relationship between $\ln(k_{ET})$ and $R_{ab}$ for MV {[Mo$_2$]–bridge–[Mo$_2$]}$^+$ complexes with the same [Mo$_2$] donor and acceptor but varying π conjugated bridges, in accordance with the decay law (Eq. (6)) in the frame of the McConnell superexchange mechanism[46],

$$k_{ET} = k_0 \exp(-\beta R_{ab}), \quad (6)$$

where $k_0$ is a kinetic prefactor and $\beta$ the attenuation factor. The linear relationship of $\ln(k_{ET})$ vs. $R_{ab}$ gives $\beta = 1.25$ Å$^{-1}$ (Supplementary Figure 9). This result indicates that the HB bridge transports the electron from donor to acceptor equally well as π bridges do, in agreement with recent reports in photoinduced ET[47].

For the three D–B–A series with different bridges, DFT calculations on the simplified models, generated by replacing the Ar groups on the formamidinate ligand with a hydrogen atoms, show that the HOMO and HOMO-1 arise from the phase-out and phase-in combinations of the δ orbitals of the two Mo$_2$ units, respectively (Supplementary Figure 10). The LUMO for 1 and 2 is the π* orbital of the bridging ligand, which permits metal(δ) to bridging ligand(π*) electronic transitions to occur[29], while for 3, similar bridge-based LUMOs are not present. Interestingly for 1 and 2, not only are the two δ-based HOMOs very similar, but the LUMOs have the same symmetry and similar density distributions, showing the similarity of the amide–amide six-membered ring to the π-conjugated phenylene group[48], despite the differences in the chemical constitution of the bridges. The HOMO–LUMO gap of 2.74 eV for 1 and 2.18 eV for 2 are in good agreement with the MLCT energies observed in the UV–Vis spectra, ca. 2.75 eV for $\mathbf{1c}^+$ and 2.55 eV for $\mathbf{2c}^+$. The δ–δ interactions in 1 and 2 are further manifested by the HOMO–HOMO-1 energy splitting of 0.09 eV (1) and 0.19 eV (2), appreciably larger than the 0.02 eV for 3. These results suggest that ET in the HB bridged Mo$_2$ dimers ($\mathbf{1}^+$) may proceed via a superexchange mechanism, specifically for $\mathbf{1b}^+$ and $\mathbf{1c}^+$ for which PT is not involved, as occurring in the π conjugated analogs ($\mathbf{2}^+$)[14].

However, simple DFT calculations show only a stationary ground state that does not take into account the fluctuational behavior of the HB bridge[47]. The small HOMO−HOMO-1 splitting for the adducts, relative to those for 2, does not fully account for the efficient mediation that the HBs have on EC and ET[26].

Significantly, this work illustrates that the electron self-exchange crossing HB interfaces can be mechanistically different in terms of PCET. For $\mathbf{1a}^+$ and $\mathbf{1d}^+$, the PCET pathway is directly evidenced by the ν(OH) stretches. In the existing PCET theories, while ET is generally described in the vibronically nonadiabatic regime, the PT process can be electronically nonadiabatic or adiabatic[8]. For the strongly coupled $\mathbf{1a}^+$, $k_{obs} < k_{adia}$ can be explained by slow PT that drags the ET process. This is the case of electronically nonadiabatic PT which takes place on a different time scale from that of ET[8]. For the strongly H-bonded $\mathbf{1d}^+$, the weak EC ($H_{ab} = 130$ cm$^{-1}$) lowers down the adiabatic ET to the PT time scale, being a concerted PCET process[7,8], with $k_{obs} \approx k_{nonadia}$. However, it appears that ET in $\mathbf{1b}^+$ and $\mathbf{1c}^+$ proceeds via a completely different ET mechanism, as evidenced by the IR spectral characteristics, specifically for the amide–amide HB moiety. While a large red-shift of the free N–H stretch is found for $\mathbf{1a}^+$ (7 cm$^{-1}$) and $\mathbf{1d}^+$ (12 cm$^{-1}$) upon one-electron oxidation (Fig. 2a, d), which signals the breakage of the bonded N–H bonds, the weak HB systems, $\mathbf{1b}^+$ and $\mathbf{1c}^+$, exhibit a small ν(NH) displacement (2 cm$^{-1}$) of the free N–H bonds (Fig. 2b, c). For all the neutral complexes, the C = O stretching band of the amide group, ν(CO), appears at 1735 cm$^{-1}$ (Supplementary Figure 11). Upon two-electron oxidation, this band is red-shifted to 1687 cm$^{-1}$ for dication $\mathbf{1c}^{2+}$, as expected for a C = O group with reduced bonding electron density. By contrast, for $\mathbf{1a}^{2+}$, which has the strongest hydrogen bonding and strongest EC, the ν(CO) band disappears completely, as a result of conversion of C = O to C–OH, induced by proton transfer, in agreement with the observation of the ν(OH) band at 3613 cm$^{-1}$ (Fig. 2a). For the two intermediate species $\mathbf{1b}^{2+}$ and $\mathbf{1d}^{2+}$, the C = O vibrational features are attenuated from $\mathbf{1b}^{2+}$ to $\mathbf{1d}^{2+}$ as the HB strength increases. All MV adducts display a weak, low energy (1687 cm$^{-1}$) C = O band, as expected. For $\mathbf{1a}^+$ and $\mathbf{1d}^+$, simultaneous appearance of ν(OH) and ν(CO) stretches in the IR spectra visualize the PCET process, as described by Fig. 2e. On the other hand, together with the absence of ν(OH), it is evidenced that in $\mathbf{1b}^+$ and $\mathbf{1c}^+$, proton transfer does not occur during the course of ET. The amide–amide central moiety functions as a delocalized six-membered ring analogous to phenylene (Fig. 2f)[48], conducting ET through the superexchange mechanism, as modeled by the DFT calculations (Supplementary Figure 10). In this case, the through-HB ET may be referred to as PUET. It is also worthwhile to note that for systems 1a and 1d, the O−H stretching bands appear in the spectra for the neutral, MV, and dicationic species (Fig. 2a, d), while all the oxidation states of 1b and 1c do not exhibit this band (Fig. 2b, c). This means that HB strength plays a critical role in the control of proton transfer, which affects the ET mechanism through interplay with the degree of EC.

To date, through-HB ET in both synthetic and naturally occurring systems has generally been treated as a PCET process[1,49,50]. This study demonstrates that ET across an HB interface may proceed via a proton-uncoupled pathway that can be more efficient than the PCET pathway because proton transfer dynamics can suppress the ET rate in PCET, as predicted by previous theories. Our results show that the PUET pathway predominates in weakly coupled systems that have weak HBs, but the mechanistic choice is subject to subtle variation of the structural and electronic factors of the system, as shown by the four HB adducts. These findings help to gain a deep and detailed understanding of ET reactions in which the redox partners are weakly electronic coupled through weak HBs, which is often the

cases in biological systems involving enzymes, proteins, and DNA[1]. For instance, in the photoactivation of DNA photolyase, sequential ET among amino-acid residues is followed by proton transfer, and thus, charge compensating simultaneous proton transfer is not a prerequisite for intraprotein radical transfer[51]. Moreover, the unprecedented results in this study should direct additional theoretical attention, from a different perspective and in a broad sense, onto ET through HBs. While PCET theories are well developed to interpret ET dynamics and kinetics through the interplay between PT and ET, efficient EC and fast ET via the PUET channel should be further elucidated in terms of semiclassical formalism, quantum mechanics, and molecular dynamics.

Collectively, we have shown that ET across an HB interface may not only proceed via a PCET pathway, as predicted by existing theories, but also by a proton-uncoupled pathway (PUET) without proton translocation, which has not been addressed theoretically or observed experimentally before. Our results indicate that ET through HBs via both PCET and PUET pathways can be as equally efficient as ET through π bonds, and superior to σ bonds. A mechanistic switch from one pathway to the other can be realized by manipulating the strengths of the HB bridge and/or the donor–acceptor EC. The findings in this study have implications for long-distance, less exothermic ET across HBs in biological systems; thus, knowledge of the proton-uncoupled pathway may lead to better understanding of biochemical processes of charge and energy transport.

## Methods

**Synthesis**. All manipulations were performed in a nitrogen-filled glove box or by using standard Schlenk-line techniques. All solvents were purified using a Vacuum Atmosphere (VAC) solvent purification system or freshly distilled over appropriate drying agents under nitrogen.

**Preparation of 1a−d**. The synthetic route is described by Supplementary Figure 12. A solution of acetamide (0.25 mmol) in 10 mL ethanol was transferred to a solution of $Mo_2(ArNCHNAr)_3(O_2CCH_3)$ (0.2 mmol) in 10 mL tetrahydrofuran (THF). After stirring at room temperature for several minutes, the volatiles were removed under reduced pressure, producing an orange yellow solid. The solid product was washed with ethanol ($3 \times 10$ mL) and collected by filtration. The dimeric structures of these compounds are confirmed by $^1$H NMR spectra in CDCl$_3$ (Supplementary Figures 13, 15, 17 and 19) and the monomeric structures by $^1$H NMR spectroscopy in DMSO−$d_6$ (Supplementary Figure 14, 16, 18 and 20).

**Preparation of 2a−d**. The synthetic route is described in Supplementary Figure 21. A solution of sodium ethoxide (0.20 mmol) in 10 mL ethanol was transferred to a solution of $Mo_2(ArNCHNAr)_3(O_2CCH_3)$ (0.20 mmol) in 20 mL THF. After stirring at room temperature for 2 h, the mixture was transferred to the terephthalic acid (0.12 mmol). After stirring for 3 h the solvents were removed at room temperature, the residue was dissolved in 15 mL of DCM and filtered through a Celite-packed funnel. The solvent was evaporated under reduced pressure. The residue was washed with ethanol ($3 \times 20$ mL) and then collected by filtration. The purity of the complexes was confirmed by $^1$H NMR spectroscopy in CDCl$_3$ (Supplementary Figures 22−25).

**Preparation of 3a−d**. The synthetic route is described in Supplementary Figure 26. A solution of sodium ethoxide (0.25 mmol) in 10 mL ethanol was transferred to a solution of $Mo_2(ArNCHNAr)_3(O_2CCCH_3)$ (0.2 mmol) in 10 mL THF. The solution was stirred at room temperature for about half an hour before the solvents were removed under vacuum. The residue was dissolved using 15 mL of DCM and filtered through a Celite-packed funnel. The DCM was removed, then 20 mL THF was added, and the solution was mixed with trans-1,4-cyclohexyldicarboxylic acid (0.12 mmol) in 10 mL ethanol. After stirring for 3 h at room temperature, the solvents were removed under vacuum. The residue was dissolved using 15 mL of DCM and filtered through a Celite-packed funnel before removing the solvent under reduced pressure. The residue was washed with ethanol ($3 \times 20$ mL) and then collected by filtration. The purity of the complexes was confirmed by $^1$H NMR spectroscopy in CDCl$_3$ (Supplementary Figures 27−30).

**X-ray structural determination**. For compounds **1a**, **1c**, and **1d**, single crystals for X-ray structure determination were obtained by diffusion of hexane into the

corresponding DCM solution and for **1b**, by the solvent (DCM) evolution method. Single-crystal data for **1a**·7CH$_2$Cl$_2$, **1b**·7CH$_2$Cl$_2$, and **1d**·CH$_2$Cl$_2$ were collected on a Rigaku XtaLAB Pro diffractometer ($\lambda = 1.54178$ Å) at 150 K and **1c** were collected on an Agilent Xcalibur Nova diffractometer with Cu-Kα radiation ($\lambda = 1.54178$ Å) at 173 K. The empirical absorption corrections were applied using spherical harmonics, implemented in the SCALE3 ABSPACK scaling algorithm[52]. The structures were solved using direct methods, which yielded the positions of all non-hydrogen atoms. Hydrogen atoms were placed in calculated positions in the final structure refinement. Structure determination and refinement were carried out using the SHELXS-2014 and SHELXL-2014 programs, respectively[53]. The solvent molecules are disordered in multiple orientations, which were refined isotropically. All non-hydrogen atoms were refined with anisotropic displacement parameters (Supplementary Table 1).

**Electrochemical studies**. Electrochemical measurements on the neutral compounds were carried out in 0.1 M $^n$Bu$_4$NPF$_6$/DCM and DMF solutions. The CVs and differential pulse voltammograms were obtained using a CH Instruments model CHI660D electrochemical analyzer with Pt working and auxiliary electrodes, an Ag/AgCl reference electrode, and a scan rate of 100 mV/s. Under these conditions, the redox potential for ferrocene, $E_{1/2}(Fc^{+/0})$, is 0.52 V. The measured potentials for the complexes are referenced to $E_{1/2}(Fc^{+/0})$.

**Spectroscopic measurements**. UV–vis–NIR and mid-infrared spectra were measured on a Shimadzu UV-3600 UV-Vis–NIR and Thermo Electron Corporation Nicolet 6700 spectrophotometer, respectively. The UV–Vis–NIR spectra were measured using an IR quartz cell with a light path length of 2 mm. The IR measurements were carried out using a thin-layer CaF$_2$ cell. The analyte concentration for the spectroscopic measurements is $1 \times 10^{-3}$ mol/L in DCM and DMF.

**Simulation of ET kinetics**. Based on the Bloch equations formalisms[54], published software Zoerbex[44] was used to simulate the coalesced free N–H vibrational bands in the spectra to derive the ET rate constants for (**1a**$^+$–**d**$^+$). Since the band shapes of the spectra are not in accord with the simple Lorentzian function, the Voigt line shapes separated into Lorentzian and Gaussian components were used to simulate the IR peaks in each spectrum (Supplementary Figures 43–46)[55]. The input parameters for simulation include the full-width at half-maximum values for each contribution to the individual peaks as well as their center frequencies and relative populations. Each N–H vibrational band in (**1a**$^+$–**d**$^+$) was analyzed by curve fitting before the Voigt simulation.

**DFT calculations**. The ORCA 2.9.1 software packages[56] were used for all DFT computations assuming an $S = 0$ spin state. DFT calculations were performed on the simplified models derived by replacing the *p*-anisyl groups in the DAniF ligands with hydrogen atoms. The geometry of the model complexes was optimized in the gas phase, employing the Becke−Perdew (BP86) functional[57,58] and RI/J approximation[59] without imposing any symmetry constraints. Geometry optimizations for the complexes were converged with the def2-SV(P) basis set[60] and def2-SVP/J auxiliary basis set[61,62] for C and H atoms, def2-TZVP(-f) basis set[63] and def2-TZVP/J auxiliary basis set[56] for N and O atoms. For Mo atoms, def2-TZVPP basis set[57] and def2-TZVPP/J auxiliary basis set[56] were used together with the ZORA approximation[64]. Tight optimization and tight self-consistent field convergence were employed along with a dense integration grid (ORCA Grid 5) for all geometry optimization calculations. Single-point calculations on optimized geometries were performed using the B3LYP functional[65–67] and the COSMO methodology[68] (using $\varepsilon = 9.08$ for DCM solvent). Isosurface plots of molecular orbitals were generated using the gOpenMol 3.00 program[69,70] with isodensity values of 0.04.

## Data availability

The X-ray crystallographic data (**1a–d**) reported in this study have been deposited at the Cambridge Crystallographic Data Centre (CCDC), under deposition number CCDC 1899032–1899035. These data can be obtained free of charge from The Cambridge Crystallographic Data Centre via www.ccdc.cam.ac.uk/data_request/cif. The data that support the findings of this study are available from the corresponding authors upon reasonable request.

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

## Acknowledgements

We acknowledge primary financial support from the National Natural Science Foundation of China (21371074), Natural Science Foundation of Guangdong Province (2018A030313894), Jinan University, and the Fundamental Research Funds for the Central Universities. We are grateful to Prof. Julia A. Weinstein (University of Sheffield) for comments and suggestions on the manuscript.

## Author contributions

C.Y.L. conceived and designed the experiments and worked on the manuscript. T.C. carried out the major experimental work and data analysis. M.M. completed the data collection and analysis of the X-ray crystal structures and S.M. performed DFT theoretical calculations. N.J.P. participated in discussion of the results and refinement of the manuscript. D.X.S., L.C., H.L.Z., S.F.Z., H.W.C., Y.Q., and Y.Y.W were involved in compound syntheses and electrochemical and spectroscopic measurements.

## Additional information

**Competing interests:** The authors declare no competing interests.

