## [Peer Review File · Nature Communications]

Reviewers' comments:

Reviewer #1 (Remarks to the Author):

This is a great study detailing the electronic effects influencing electron transfer across hydrogen bonded mixed valence complexes and their comparisons to analogous systems where electron transfer is mediated by bridges based upon sigma and pi bonds. The study is one of the most detailed to date that address mechanistic questions regarding how hydrogen-bonds mediate electron transfer in mixed valence complexes. The experimental data are clean, elegant, and well presented.

1. While it is clear that complexes 1a–1d are completely dimerized in the neutral states, it is not clear how oxidation affects the dimerization equilibria. The authors note that upon oxidation “multiple $\nu(\text{NH})$ bands are observed highlighting the multiple nuclear degrees of freedom due to ET or PCET...” could some of these bands arise from monomeric species present? i.e. do the dimers stay intact in the singly and doubly oxidized states or are monomers also present?
2. For complex 1a+ it is quite surprising that such a strong IVCT band is observed for a complex that involves a competitive PCET pathway. Even more surprising is that an IVCT band is observable at all for complexes 1a and 1d given the fact that an $\nu(\text{OH})$ band is observed in all oxidation states. The point is that the presence of both $\nu(\text{OH})$ and $\nu(\text{CO})$ bands for the same complex indicates equilibrium. That equilibrium obviously would require a large nuclear displacement. The observation of an intense IVCT band is not consistent with a large nuclear displacement.
3. For complexes 1a+–1d+ considering that ET is predicted to occur on a timescale near that of the EPR timescale it is quite surprising that only a spectrum consistent with a localized electron on Mo is observed, and that no involvement of the ligands/bridging ligands are observed.
4. One of the most interesting aspects of this system is that ET occurs across a hydrogen bonded bridge where there seems to be almost no evidence of significant charge density (from their DFT and EPR results) on that bridge. The lack of significant electronic involvement of the bridging ligand is further suggested by the author’s observation of only one Gaussian shaped IVCT band in the nIR for these complexes. These findings are contrary to those observed by Kubiak where electron transfer across hydrogen bonds and pi-pi interactions was found to have a direct relation to the degree of electron-spin density residing on the bridging ligands. This begs the question about the appropriate model for ET. (1) Is the hydrogen-bond directly supporting ET by allowing charge to flow through it, or is it dynamic in nature such that the hydrogen bond interaction allows for the donor-acceptor sites to occasionally come into close proximity allowing the electron to “hop” through space.
5. For supplementary Figure 11, what happens to the $\nu(\text{CO})$ band in the singly-oxidized mixed valent state, only the isovalent neutral and doubly oxidized states are shown.

Minor corrections:

6. Page 5, line 17, consider the use of a table to present the structural data.
7. Several typos are present throughout the manuscript, most notably is that numerous kobs are incorrectly typed as kabs. Suggestions for future study:
8. The inclusion of crystal structures for all of the hydrogen bonded complexes in their neutral state is a beautiful addition to this study. It would be very interesting however if crystal structures of the oxidized states could also be obtained, this would allow a direct calculation of the reorganization energies involved in these structures.

Comments on NCOMMS-18-33721-T:

This work titled as “Efficient electron transfer across hydrogen bond interface: proton-coupled versus uncoupled pathways” by Liu and his coworkers described that electron transfer through hydrogen-Bonds could proceed via not only PCET pathway but also PUET pathway using covalent-bonded Mo₂ dimer with bridged amide-amide dual hydrogen-bonds as model complex. This work is definitely interesting work in the field of electron transfer of IVCT complexes. However, in my personal opinion, this work is still preliminary and the following issues should be solved before this work to be considered for publication on *Nature Communication*.

1. The potential in organic solvent was better be reported as vs Fc⁺⁰ (adding ferrocene as internal reference), because Ag/AgCl is not common used as reference electrode for organic solvent.
2. For both **1a** and **2a** with NMe₂ group are special cases compared with other complexes according to the data summarized in Table 1. Please give more explanation.
3. For the mechanistic analysis of PCET and PUET, the current experiments are not fully convincing. (1) if using D-atom instead the H-atom, whether the IVCT band will affect by deuterium for PCET and PUET, respectively? (2) How about the kinetic isotope effect (k_H/k_D) for PCET and PUET respectively? (3) How the Hydrogen-bond strength change with different X-substituent group on ligand? And further, what's the relationship between the k_{PCET} (or k_{PUET}) and the Hydrogen-bond strength? (4) Actually, the proton transfer “effective distance” in PCET should related with the Hydrogen-bond structure and the strength, I suggest the author giving more insight on this system, rather than only with simple DFT analysis.

Responses to Referee

(NCOMMS-18-33721-T)

Authors: We thank the Reviewers' very much for their supportive and informative comments. The manuscript is revised based on reviewers' comments in hope of addressing their concerns. Here, we response the reviewers' comments line by line to clarify the concerned issues or answer the questions raised.

Reviewer #1 (Remarks to the Author):

This is a great study detailing the electronic effects influencing electron transfer across hydrogen bonded mixed valence complexes and their comparisons to analogous systems where electron transfer is mediated by bridges based upon sigma and pi bonds. The study is one of the most detailed to date that address mechanistic questions regarding how hydrogen-bonds mediate electron transfer in mixed valence complexes. The experimental data are clean, elegant, and well presented.

1. While it is clear that complexes 1a–1d are completely dimerized in the neutral states, it is not clear how oxidation affects the dimerization equilibria. The authors note that upon oxidation “multiple $\nu(\text{NH})$ bands are observed highlighting the multiple nuclear degrees of freedom due to ET or PCET...” could some of these bands arise from monomeric species present? i.e. do the dimers stay intact in the singly and doubly oxidized states or are monomers also present?

Authors: It is unambiguous that Mo₂ complexes 1a-1d in different oxidation states (neutral, singly and doubly oxidized) exist as hydrogen bonded dimers in less polar solvents (such as DCM). Oxidation of the redox centers (here Mo₂, which has relatively low potential) usually help dimerization through hydrogen bond; in other words, the hydrogen bonding in the oxidized dimers should be stronger because H-bond is electrostatic interaction in nature. For example, Kubiak (J. Am. Chem. Soc. 2018, 140, 12756–12759, Chem. Sci., 2017, 8, 7324-7329) and others found that in their systems, the building block exists as monomer in neutral and oxidization initiates the dimerization, as indicated by the electrochemical analyses. In this study, ¹H NMR spectra indicate that the Mo₂ complexes are dimerized in neutral, and all the experimental results indicate that the oxidized complexes are H-bonded dimers in solution. For example, electrochemistry shows two unresolved redox couples, but monomer should give single redox couple. The IVCT band for each of the complexes, which is comparable to the covalent bond analogues with respect to energy and band shape, is a direct evidence for the H-bonded dimers because singly oxidized monomer does not exhibit an IVCT band. In this series all the complexes have one redox center, Mo₂, in the electrochemically measured range, except for the 1a which has the redox active substituents N(CH₃)₂ with a much higher potential ($E_{1/2} > 0.6$ V Ag/AgCl). Therefore, chemical oxidation using FcPF₆ ($E_{1/2} \sim 0.5$ V) only oxidation of Mo₂⁴⁺ to Mo₂⁵⁺ ($E_{1/2} \sim 0.3$ V). Singly oxidized complexes from **1a**⁺ to **1d**⁺ exhibit different spectra in both near-IR (IVCT) and IR (vibrational modes) regions, which

represent the distinct mixed-valence properties for the dimers. For the doubly oxidized complexes the IVCT bands disappear, just like covalent bond bridged Mo₂ dimers.

In the revised text, we have added more discussion to address the dimeric structure and the related features for the oxidized species in the related context. The added lines are highlighted

Reviewer #1

2. For complex 1a⁺ it is quite surprising that such a strong IVCT band is observed for a complex that involves a competitive PCET pathway. Even more surprising is that an IVCT band is observable at all for complexes 1a and 1d given the fact that a $\nu(\text{OH})$ band is observed in all oxidation states. The point is that the presence of both $\nu(\text{OH})$ and $\nu(\text{CO})$ bands for the same complex indicates equilibrium. That equilibrium obviously would require a large nuclear displacement. The observation of an intense IVCT band is not consistent with a large nuclear displacement.

Authors: We agree with Reviewer 1; indeed, this study raises some important questions about electron transfer through hydrogen bonds. For the neutral 1a and 1d, we do observe $\nu(\text{OH})$ and $\nu(\text{CO})$ modes simultaneously. This indicates that there is equilibrium between $\text{N}-\text{H}\cdots\text{O}$ and $\text{N}\cdots\text{H}-\text{O}$ bridged species. This automerization process is probed reasonably by fast time scale IR. But for the mixed-valence dimers (1a⁺ and 1d⁺), the $\nu(\text{CO})$ band at 1680 cm^{-1} is very weak (Figure S11 in the revised SI) compared to those for 1b⁺ and 1c⁺, because the equilibrium is now controlled by electron transfer, in accordance with the proton-coupled electron transfer. It appears that the H-bonded bridge functions like a conjugated π bridge that allows superexchange to occur, even in the case the protons are in motion. This hypothesis is supported by DFT calculation. But still, observation of IVCT band in PCET systems is very interesting and phenomenal.

Reviewer #1

3. For complexes 1a⁺–1d⁺ considering that ET is predicted to occur on a timescale near that of the EPR timescale it is quite surprising that only a spectrum consistent with a localized electron on Mo is observed, and that no involvement of the ligands/bridging ligands are observed.
4. One of the most interesting aspects of this system is that ET occurs across a hydrogen bonded bridge where there seems to be almost no evidence of significant charge density (from their DFT and EPR results) on that bridge. The lack of significant electronic involvement of the bridging ligand is further suggested by the author's observation of only one Gaussian shaped IVCT band in the nIR for these complexes. These findings are contrary to those observed by Kubiak where electron transfer across hydrogen bonds and π - π interactions was found to have a direct relation to the degree of electron-spin density residing on the bridging ligands. This begs the question about the appropriate model for ET. (1) Is the hydrogen-bond directly supporting ET by allowing charge to flow through it, or is it dynamic in nature such that the hydrogen bond interaction allows for the donor-acceptor sites to occasionally come into close proximity allowing the electron

to “hop” through space.

Authors: Comments 3 and 4 and related. The electronic structures for the Mo₂-Mo₂ dimers (with covalent bond or H-bond bridges) are clear and only the delta electrons and the bridging ligand are involved in the charge transfer process. The unique electronic structure of Mo₂ complex renders the Mo₂-Mo₂ MV system very clear electronic transition spectra and vibronic IVCT band. From the relatively high IVCT band energy and broad, Gaussian-shaped band profile, these H-bond adducts are in the valence-trapped Class II regime in Robin-Day's scheme, which allows observation of behaviors of localized Mo₂ in near-IR or IR. For strongly coupled, pi bridged Mo₂ dimers, we usually observe MLCT for neutral, and MLCT with LMCT for the mixed-valence. For these weakly coupled H-bond dimers, charge transfer between Mo₂ and bridging ligand (MLCT) occurs at high energy and in a fast timescale close to that of spectroscopic technique, separated from the IVCT even for weakly coupled systems. This is why the Mo₂-Mo₂ system is particularly suitable for optical analysis under the two-state model framework. Concerning the model and mechanism for ET crossing H-bond, we tend to believe that molecular dynamics resulting from H-bond flexibility makes significant contribution in the ET pathways. But for the time being, we don't have ensured answers for such questions. It seems that ET across H-bond is still unpredictable. There is much unknown worthy of exploring.

5. For supplementary Figure 11, what happens to the nu(CO) band in the singly-oxidized mixed valent state, only the isovalent neutral and doubly oxidized states are shown.

Authors: In revised supplementary material, Figure S11 is modified by adding the spectrum for the mixed-valence adduct, which shows the variation of the nu(CO) band for the oxidation state change from 0 to 2+ via 1+. Again, we see the optical behaviors of 1a+ and 1d+ different from those of 1b+ and 1c+. For 1b+ and 1c+, particularly for 1c+, the nu(CO) band is shifted to 1687 cm⁻¹ from 1735 cm⁻¹ and increases intensity upon removal of two electrons. For 1a+ and 1d+, this band is weak and disappears for 1a²⁺ and 1d²⁺. These results collectively support that in 1a and 1d, the bridging protons are in motion, but 1b and 1c show different story---the bridging protons do not move while electron transfer occurs. So, these are the direct evidences for proton-coupled and uncoupled electron transfer.

Minor corrections:

6. Page 5, line 17, consider the use of a table to present the structural data.

Authors: Table of selected bond distance and angles is added

7. Several typos are present throughout the manuscript, most notably is that numerous kobs are incorrectly typed as kabs. Suggestions for future study:

Author: Typos are corrected

8. The inclusion of crystal structures for all of the hydrogen bonded complexes in their neutral state is a beautiful addition to this study. It would be very interesting however if crystal structures of the oxidized states could also be obtained, this would allow a direct calculation of the reorganization energies involved in these structures.

Author: Growing single crystals of the mixed-valence compound for structural analyses is very difficult, especially for those in Class II. Presumably, it is because the counter ion is in motion due to electron transfer. Although efforts have been made for years, so far only fully localized (J. Am. Chem. Soc. 2003, 125, 12945-12952) and fully delocalized (J. Am. Chem. Soc. 2004, 126, 14822-14831) mixed-valence Mo₂ dimers are structurally characterized by Cotton's group.

Reviewer #2 (Remarks to the Author):

This work titled as "Efficient electron transfer across hydrogen bond interface: proton-coupled versus uncoupled pathways" by Liu and his coworkers described that electron transfer through hydrogen-bonds could proceed via not only PCET pathway but also PUET pathway using covalent-bonded Mo₂ dimer with bridged amide-amide dual hydrogen-bonds as model complex. This work is definitely interesting work in the field of electron transfer of IVCT complexes. However, in my personal opinion, this work is still preliminary and the following issues should be solved before this work to be considered for publication on Nature Communication.

1. The potential in organic solvent was better be reported as vs Fc^{+/0} (adding ferrocene as internal reference), because Ag/AgCl is not common used as reference electrode for organic solvent.

Authors: This is done in the revised manuscript. We determined the potential for in the same electrochemical system and conditions as for the sample measurements. In this Ag/AgCl system, E_{1/2}(Fc^{+/0}) is 0.52 V. In the revised manuscript, all the electrochemical potentials are converted to volt vs. Fc^{+/0}. We did not carry out the experiments using ferrocene internal reference because the potentials are in the similar range and the CVs are overlapped.

2. For both 1a and 2a with NMe₂ group are special cases compared with other complexes according to the data summarized in Table 1. Please give more explanation.

Authors: In the revised manuscript, more explanation is given to the electrochemistry and spectroscopy for 1a and 2a. See the highlighted lines on page 9 and 10.

3. For the mechanistic analysis of PCET and PUET, the current experiments are not fully convincing. (1) if using D-atom instead the H-atom, whether the IVCT band will affect by deuterium for PCET and PUET, respectively? (2) How about the kinetic isotope effect (k_H/k_D) for PCET and PUET respectively? (3) How the Hydrogen-bond strength change with different X-substituent group on ligand? And further, what's the relationship between the k_{PCET} (or k_{PUET}) and the Hydrogen-bond strength? (4) Actually, the proton transfer "effective distance" in PCET should related with the Hydrogen-bond structure and the strength, I suggest the author giving more insight on this system, rather than only with simple DFT analysis.

Authors: For (1) and (2). Since the manuscript was submitted, we have been making efforts for to carry out the study on D/H isotope effect on ET dynamics and kinetics. We have been experiencing difficulties in making pure D-complexes and measuring the deuterated samples without D/H exchange in solution. Unfortunately, we do not have data for this manuscript revision.

(3) In the revised text, we have presented the structural parameters for the hydrogen bonded central moiety in the added Table 1, and a brief discussion is given. The compounds are named as a, b, c and d in the order of increasing the electron donating of the X substituents (N(CH₃)₂, C(CH₃)₂, OCH₃ and CH₃) and defined by the Hammett constants (σ_x), but the structural parameters and spectroscopic data indicate that the order of hydrogen bond strength is 1a > 1d > 1b > 1c, indicating more complicated factors that may affect the hydrogen bond strength. Yes, the two strong hydrogen bonded systems, 1a⁺ and 1d⁺, undergo PCET, while in the weakly boned 1b⁺ and 1c⁺ electron transfer proceeds via PUET.

(4) In this work, only for 1a⁺ and 1d⁺, proton transfer occurs. In the structures, we see shorter H···O distances for these two adducts, which defines the stronger hydrogen bonds for them. Strong hydrogen bonds lower the proton transfer barrier so that proton transfer occurs. Therefore, the structure data are consistent with the spectroscopic results and support the PCET theory. It seems to us that molecular dynamics makes significant contribution to electron transfer in hydrogen bond systems in both PCET and PUET pathways. We are developing and studying a new system in hope of giving further and deep elucidation to the molecular dynamic effect on through hydrogen bond electron transfer.

REVIEWERS' COMMENTS:

Reviewer #1 (Remarks to the Author):

After reviewing the revised manuscript, I am satisfied with the authors responses to our concerns. There is no doubt that the study is novel and begins to answer some very fundamental questions about ET across H-bonds. The only minor correction I would add is with respect to comment #1. Please see the summary at the end of these comments.

Overall, I think the study was high impact and highly detailed in addressing some of the more intimate details of hydrogen bonded ET vs PCET. As with most studies, while some questions were answered, several more presented themselves. One of the most interesting aspects is the observations of IVCT bands for all dimers despite an apparent proton transfer for two of the complexes. The authors rebuttal indicating that the monomeric species do not display IVCT bands supports that ET is occurring across the hydrogen bond in a dimer and their attempt to explain this through a PUET pathway is credible. While more data is definitely needed to better address this idea, the communication format is perfect for the current results and highly thought provoking. I also agree with reviewer 2 who noted that KIE studies are needed, however I agree with the authors who wish to defer those questions to a followup manuscript. That certainly seems appropriate. There will be much work in the future on the issues raised in the communication.

Original Reviewer #1 Comments, Author Response, 2nd Reviewer Response below:

1. While it is clear that complexes 1a–1d are completely dimerized in the neutral states, it is not clear how oxidation affects the dimerization equilibria. The authors note that upon oxidation “multiple $\nu(\text{NH})$ bands are observed highlighting the multiple nuclear degrees of freedom due to ET or PCET...” could some of these bands arise from monomeric species present? i.e. do the dimers stay intact in the singly and doubly oxidized states or are monomers also present?

Authors: It is unambiguous that Mo₂ complexes 1a-1d in different oxidation states (neutral, singly and doubly oxidized) exist as hydrogen bonded dimers in less polar solvents (such as DCM). Oxidation of the redox centers (here Mo₂, which has relatively low potential) usually help dimerization through hydrogen bond; in other words, the hydrogen bonding in the oxidized dimers should be stronger because H-bond is electrostatic interaction in nature. For example, Kubiak (J. Am. Chem. Soc. 2018, 140, 12756–12759, Chem. Sci., 2017, 8, 7324-7329) and others found that in their

systems, the building block exists as monomer in neutral and oxidization initiates the dimerization, as indicated by the electrochemical analyses. In this study, ^1H NMR spectra indicate that the Mo_2 complexes are dimerized in neutral, and all the experimental results indicate that the oxidized complexes are H-bonded dimers in solution. For example, electrochemistry shows two unresolved redox couples, but monomer should give single redox couple. The IVCT band for each of the complexes, which is comparable to the covalent bond analogues with respect to energy and band shape, is a direct evidence for the H-bonded dimers because singly oxidized monomer does not exhibit an IVCT band. In this series all the complexes have one redox center, Mo_2 , in the electrochemically measured range, except for the 1a which has the redox active substituents $\text{N}(\text{CH}_3)_2$ with a much higher potential ($E_{1/2} > 0.6 \text{ V Ag/AgCl}$). Therefore, chemical oxidation using FcPF_6 ($E_{1/2} \sim 0.5 \text{ V}$) only oxidation of Mo_2^{4+} to Mo_2^{5+} ($E_{1/2} \sim 0.3 \text{ V}$). Singly oxidized complexes from 1a⁺ to 1d⁺ exhibit different spectra in both near-IR (IVCT) and IR (vibrational modes) regions, which represent the distinct mixed-valence properties for the dimers. For the doubly oxidized complexes the IVCT bands disappear, just like covalent bond bridged Mo_2 dimers. In the revised text, we have added more discussion to address the dimeric structure and the related features for the oxidized species in the related context. The added lines are highlighted

Reviewer 1: Thank you for clarifying these concerns and adding additional text to the manuscript. While the electrochemical data does support the dimerized structure in DCM, the observation of nIR bands is much more impactful. Consider addition of a discussion on how the singly oxidized monomers do not exhibit an IVCT band. This directly supports the dimerized structure for 1a⁺–1d⁺ as the appearance of the IVCT bands is solely due to dimerization.

Reviewer #2 (Remarks to the Author):

In this updated manuscript, the authors have give the response on most of the issues provided by referees, and this work give a elegant case for research on mechanistic questions regarding how hydrogen-bonds mediate electron transfer in mixed valence complexes. Despite the author do not fully solved the KIE experiments to support the mechanistic studies in this stage, I still pleased to suggest this work to be published on Nature Comm..

Responses to Referees

Authors: We sincerely thank the reviewers for their valuable comments on the manuscript and for their support for publication of the manuscript in Nature Communication. Indeed, as pointed out by both reviewers, with these reported preliminary results and conclusion there is much to do to substantiate and extend the study in this field.

Reviewer 1: Thank you for clarifying these concerns and adding additional text to the manuscript. While the electrochemical data does support the dimerized structure in DCM, the observation of NIR bands is much more impactful. Consider addition of a discussion on how the singly oxidized monomers do not exhibit an IVCT band. This directly supports the dimerized structure for $1a^{+}-1d^{+}$ as the appearance of the IVCT bands is solely due to dimerization.

Authors : Reviewer 1 suggests to give additional discussion on the UV-Vis-NIR spectra of the singly oxidized Mo_2 monomers, in comparison with the IVCT absorptions for the adducts, to further support the hydrogen bonded dimeric structures of the singly oxidized complexes. We agree and appreciate this suggestion. In the revised manuscript, a discussion of several lines is added (page 10, yellow highlighted) to the related contents. More refs are cited to address the IVCT band features in various Mo_2 dimers with covalently bonded bridges. The discussion refers to the spectra of the complexes in polar solvent (DMF) in which the molecules exist in monomer. In the spectra (supplementary

Figure 38), there is no any absorption observed in the NIR region. Text of this discussion is quoted here:

“For both of the systems, the IVCT band parameters, including transition energy (E_{IT}), band intensity (ϵ_{IT}) and half bandwidth ($\Delta\nu_{1/2}$), are comparable with those found for other Mo_2 dimers with various π bridges differing in length,**Error! Bookmark not defined.**^{31,32} symmetry,**Error! Bookmark not defined.** conjugation³³ and conformation.³⁴ As expected, the absorption spectra of the corresponding monomers, $[\text{Mo}_2(\text{DArF})_3(\text{O}_2\text{CCONH}_2)]^{0/+}$, obtained in DMF do not exhibit the characteristic IVCT transition, as shown in Supplementary Figure 38. Therefore, the presence of a well-defined IVCT band for each of the HB adducts confirms their dimeric structure and the mixed-valency of the singly oxidized complexes.”